

# 1 From Virtual Field Trip to Geologically-Reasoned Decisions in Yosemite Valley

Nicolas C. Barth[1*], Greg M. Stock[2], Kinnari Atit[3]
[1] Department of Earth & Planetary Sciences, University of California, Riverside, CA 92521
[2] National Park Service, Yosemite National Park, El Portal, CA 95389
[3] Graduate School of Education, University of California, Riverside, CA 92521
* Corresponding Author: nic.barth@ucr.edu

## 7 ABSTRACT

This study highlights a Geology of Yosemite Valley virtual field trip (VFT) and companion exercises produced as a
four-part module to substitute for physical field experiences. The VFT is created as an Earth project in Google Earth
Web, a versatile format that allows access through a web browser or Google Earth app with the sharing of an internet
address. Many dynamic resources can be used for VFT stops through use of the Google Earth Engine (global satellite
imagery draped on topography, 360° street-level imagery, user-submitted 360° photospheres). Images, figures, videos,
and narration can be embedded into VFT stops. Hyperlinks allow for a wide range of external resources to be
incorporated; optional background resources help reduce the knowledge gap between general public and upper-
division students, ensuring VFTs can be broadly accessible. Like many in-person field trips, there is a script with
learning goals for each stop, but also an opportunity to learn through exploration as the viewer can dynamically change
their vantage at each stop (i.e. guided discovery learning). This interactive VFT format scaffolds students' spatial
skills and encourages attention to be focused on a stop's critical spatial information. The progression from VFT to
mapping exercise to geologically-reasoned decision-making results in high quality student work; students find it
engaging, enjoyable, and educational.

## 21 1. INTRODUCTION

The shifting landscape of the global COVID-19 pandemic in early 2020 brought unprecedented uncertainty and
disruption to educators worldwide, particularly field educators. To promote safety and minimize virus spread, many
national agencies, local governments, and universities changed rules and guidelines on a near-weekly basis, often
implementing drastic procedural changes with little notice. Against this backdrop a hundred-plus intensive upper-
division field geology courses (i.e. "summer field") hosted by universities around the world that were scheduled to
run over the summer were forced to make a hard choice: Do we proceed? Some instructors held out hope of running
their course only to have their camping permits denied at the last minute or shifting regulations cancel their field
course altogether. Although a very small minority did actually run in-person courses with clearances and modifications
to safely limit COVID-19 exposure, most instructors shifted to offering alternative courses in a remote or virtual
format that would attempt to instill some of the field skills considered essential by employers and graduate advisors.
Field geology educators across the globe began organizing and meeting virtually in March 2020 to assess resources
readily available for use in remote field geology courses, what could be adapted from instructors' courses, and what
could be created new with enough lead time to be implemented by others. Several working groups coordinated by





affiliates of the National Association of Geoscience Teachers (NAGT) sprung up with sub-interests (learning
objectives, building a community-based virtual field camp, virtual field trips, virtual worlds, virtual field geophysics,
etc.). A clear challenge that repeatedly arose was how to virtually deliver the same depth of learning that in-person
field experiences provide.
For the preceding three years University of California- Riverside's summer field geology course, hosted in Owens
Valley, California, had a short one-day module where students were expected to use web-hosted lidar data to map
geomorphic features such as debris fans, rockfalls, and glacial moraines in Yosemite Valley. This exercise was
followed by a long day of driving over the Sierra Nevada to Yosemite Valley where students would participate in a
multi-stop walk highlighting natural hazards including floods, rockfalls, and rock avalanches. In post-course surveys
students cited the module and visit as one of their favorite aspects of the course, and the instructor thought the mapping
products students produced in this module were some of their best in the course. Out of this background familiarity
and potential for a long-lived module (potential for broad international interest and usable with a site visit once in-
person field instruction resumed), we developed a four-part Geology of Yosemite Valley module, advertised through
the main NAGT working group, and made this module publicly available in May 2020 through the Science Education
Resources Center (SERC) website hosted by Carleton College (Figure 1). This paper is intended to provide an
overview of the learning philosophies and technologies employed, with the hope that it promotes the creation of high
quality virtual field trips (VFTs) appropriate for the general public, and highlights how VFTs can lead into advanced
level mapping and geologically-reasoned decision-making exercises suitable for upper-division courses.

## 2. MODULE DESCRIPTION

Geology of Yosemite Valley is a four-part module that we intentionally designed to be flexible in duration and student
grade-level (https://serc.carleton.edu/NAGTWorkshops/online_field/ activities/237092.html; last access: 24 August
2021). Full functionality requires a computing device with internet connection, Google Chrome browser, Google Earth
Pro desktop, and a trackpad or mouse. The four parts are described in the suggested order of completion, as each part
builds on the previous one. Particular detail is given to the virtual field trip (Part I) created through the Projects feature
of Google Earth Web, a user-friendly and highly-adaptable format with advantages over many other VFT platforms.
An overview of Parts II, III, and IV are provided to illustrate how a VFT can be used as background material for more
advanced-level mapping and writing exercises.

### 2.1 Geology of Yosemite Valley Virtual Field Trip

2.1.1 Introduction to Google Earth Web Projects

The Geology of Yosemite Valley VFT was created as an Earth project using the *Creation Tools* available in Google
Earth Web accessible via web browser. Google introduced *Tour Builder* as a beta tool in 2013 as a way to create
shareable place-based narratives with images, text, videos and links hosted within their Google Earth engine; *Tour*
*Builder* was discontinued in July 2021, but most of the functionality has been made permanent in Google Earth Web's
*Creation Tools*, which launched in 2019. Creating with *Creation Tools* requires signing into the web version of Google





Earth (preferred browser: Google Chrome) with a Google account (free); all projects are stored in the creator's Google
Drive storage system. Much of the functionality will be familiar to users of other Google products (Docs, Sheets,
Slides, etc.): a creator can make a project shareable or completely private, collaborate with other creators if they
choose, and project changes are immediately saved and updated with an internet connection (Figure 2).
All created features for a given project appear in a Table of Contents and can be reordered by the creator but not
viewers. A publicly-shared project can be downloaded by a third-party user; it can also be copied such that a third-
party creator can edit and adapt someone's project as their own (for example adding their own built-in quiz
assessments for their students).
Three types of features can currently be added to projects: fullscreen slides, lines or shapes, and placemarks.
(1) Fullscreen slides fill the entire screen and are not geotagged. The creator can upload a background image or

video and overlay text with hyperlinks. Because of the lack of geotagging, fullscreen slides work best as title

slides or interstitial slides that intentionally pull focus away from the more interactive Google Earth

environment (for example to highlight a figure or concept).

(2) Placemarks are geotagged points in Google Earth's global environment and are the most versatile feature that

can be created. What makes placemarks truly standout is that not only is the viewer flown to the point on the

globe, but the precise view (zoom level, look direction, imagery) that the creator selects. The view can be a

top-down or oblique 3D vantage of Google Earth's elevation model-draped satellite imagery or pulled from

Google Street View's extensive ground-level photosphere imagery (outward looking 360° zoomable

imagery). User-submitted photosphere imagery can also be incorporated if the use is consistent with their

Creative Commons license. VFT creators can use the Google Street View app on a smartphone or tablet to

create and submit their own photospheres, which can be readily used as well. The creator can add text,

hyperlinks, images, and videos into a sidebar that helps explain the particular view selected for the placemark.

There is a more advanced option to directly edit the sidebar's html, which allows for custom widths, styles,

and the addition of features like audio narration or embedded quizzes.

(3) Lines or polygons can be drawn on the standard 3D view of Google Earth with different widths, colors and

transparencies, which may be helpful for highlighting a specific feature like a fault or landslide. Similar to

the placemark functionality, a sidebar box can have text or images describing the feature and a custom view

can be tagged. Each line appears as an item in the Tables of Contents so this would be an unwieldy way to

annotate many features in a small area (creating and embedding a figure would be more effective). It is not

possible to create lines or polygons within a photosphere view.

## 2.1.2 VFT Walkthrough

When the viewer (student) clicks on the provided Geology of Yosemite Valley weblink on a computer, the default
browser opens an uneditable "view only" version of the project (VFT) in Google Earth Web. On a tablet or smartphone
the Google Earth app is automatically opened if downloaded. The format dynamically adjusts to the size of the screen;
larger screens are able to convey more information and view at once and are probably ideal in most situations.



A Table of Contents appears with all of the tour items and a selectable "Present" button. Clicking or tapping this
button takes you to the first item (i.e. first virtual field trip stop) in the Table of Contents and starts what is effectively
an interactive slideshow. This VFT moves from the past towards the present, starting with a general overview of
Cretaceous geology, bedrock joints, glaciations, and then moves on to active processes like rockfalls and debris fans.
The VFT is designed in a format that could be a standalone overview of Yosemite Valley geology, but has the second
purpose of preparing the student for additional exercises on geomorphic mapping; relatedly there is a particular
emphasis on Quaternary deposits and hazards in the VFT.
At any point, navigation allows the student to go forward or backward or pull up the Table of Contents to revisit a
past stop. To maximize accessibility to a wide variety of knowledge levels, text in the VFT is extensively hyperlinked
to external web resources like Wikipedia or the U.S. Geological Survey. If a student is familiar with terms such as
"subduction zones" or "partial melting" then they can read on, whereas another student who is not familiar with those
terms readily has access to the information needed to understand. A student works through each of the 44 stops,
reading the text and connecting the views to annotated figures and videos provided. In some places small exercises
are suggested using the built-in measure tool (measuring area or distance). A student that needs to dive deeper into
the hyperlinks and background information will need more time to complete the VFT (~3 hours), while a more
knowledgeable student may be able to work through it in under 2 hours.
Yosemite Valley was a fortunate place to design a VFT, as the heavy visitation and high interest meant that many
existing resources could be adapted rather than being created new. Google Earth's photogrammetry-derived elevation
model-draped satellite imagery in Yosemite Valley is of exceptional quality, on par with coverage in major cities. The
National Park Service has a wide variety of professionally-produced informative YouTube videos on Yosemite,
including many on geology topics, that we incorporated with attribution. Similarly, most of the valley's main trails
and viewpoints have Google Street View coverage. We created many new figures to best pair with specific vantages
(Figure 3).
To be most broadly applicable, we designed this VFT without any assessment. Students get participation credit for
completing the VFT; if they are less diligent or skip stops they are more likely to need to revisit the VFT later to
complete the other module parts. Teachers could create their own quizzes hyperlinked at different stops, even linking
them directly to their gradebook if they had that functionality.
While the VFT could be conducted as a guided tour with an instructor leading the class through the field trip in a
lecture format (this would probably take less time), the simplicity of the format and opportunities for deeper learning
are greater if students are allowed to guide themselves (1-2 hours). Like any good in-person field trip, the Google
Earth Web project format strikes an excellent balance between purpose-driven stops with targeted learning goals and
opportunities for learning through exploration. While the particular vantages, text, images, and videos are presented
in a structured format to deliberately guide a student, at any point a student can break the script by zooming in, rotating
their view, walking a trail, or even searching for additional photospheres. It is a simple matter of continuing where
they left off in the Table of Contents. The mixture of medias and vantages allows for a particularly dynamic and





engaging format. Peer reviewers, students, and general public viewers all praised the VFT format. Creating a VFT
using Google Earth Web *Creation Tools* is intuitive and easily taught; students could create their own VFT as an
alternative to an oral slide-based presentation (e.g. Senger et al., 2021).

## 2.2 El Capitan Cross-Cutting Relationships Exercise

Part II of the module leverages the novel Geologic Map of the Southeast Face of El Capitan (Putnam et al., 2014;
2015) as an advanced real-world relative dating exercise. El Capitan is one of the most famous landmarks in Yosemite,
an imposing, nearly kilometer-high vertical-walled monolith of exposed intrusive rock steeped in the history of rock
climbing. With the help of rock climbers and gigapixel photographs, geologists mapped intrusive units with 8 different
chemical compositions projected onto a vertical plane with great detail and precision (10 cm accuracy for most
contacts). While some of the units may be time-transgressive (e.g. multiple generations of pegmatite dikes) or coeval
(overlapping geochronologic ages and gradational contacts), relationships are consistent enough that students can
screenshot a particular region of the map and interpret the relationships to sort the 8 units from oldest to youngest
(Figure 4). An instructor-only file is provided with verified credentials.
With author permission, a version of the geologic map (Putnam et al., 2014) was edited to remove details such as the
geologic summary, correlation of mapped units, and geochronology so that students could focus on the mapped
relationships. An El Capitan stop on the VFT provides the necessary background that students need to understand the
basics of this geologic puzzle exercise. The concepts of cross-cutting relationships, gradational contacts, included
fragments, and magma mixing are introduced, using relatable examples where possible (a cracked phone screen for
cross-cutting relationships) and students are provided an embedded link to download the modified map. Because
students could search for the geologic map and companion journal article, this exercise works best as an in-session
group exercise with lenient grading. Emphasis should be placed more on the locations the students decide to
screengrab to highlight the least ambiguous relationships and their explanations provided.
Many cross-cutting relationship exercises given to students may involve sorting the timing of layered units,
unconformities, folds, faults, and dikes. This El Capitan exercise is made more challenging by focusing on eight
intrusive units with limited spatial layering and real world relationships of varying ambiguity.

## 2.3 Geomorphic Mapping of Yosemite Valley Exercise

In Part III of the module students are tasked with producing a geomorphic map of the Quaternary (mostly Holocene)
surficial deposits in Yosemite Valley using Google Earth Pro (desktop version)(Figure 5). Features to be mapped
include deposits of talus (i.e., rockfall), debris fans, rock avalanches, and glacial moraines, as well as river terrace
risers. The VFT highlights examples of each of these deposit types with additional background concerning processes
and linked hazards (rockfall, flooding, debris flows), including historic events. All the instructions and data links
needed for this mapping exercise are contained within the Google Earth KMZ provided to students. An example of
each deposit type or feature to be mapped is provided to students in the folder structure that they will use to submit
their final map (as a KMZ). Students create their map using web-served high-resolution hillshades derived from



unfiltered (i.e. includes trees and buildings) and last return (i.e. bare-earth) airborne lidar data and the internal web-
served Google Earth satellite imagery built into the Google Earth engine. The different Quaternary features (debris
fans, rock avalanches, etc.) have very distinct textural and slope styles readily distinguishable in the bare-earth lidar
hillshade; once students gain an eye for it they can fairly efficiently map the 75 km$^2$ Yosemite Valley region.
This mapping exercise is most appropriate for an upper-division course (e.g. geomorphology, applied geology,
summer field). The maps are graded based on correctness, completeness, and neatness; students are held to a
professional standard. An entire class' KMZ maps can be added to the instructor's Google Earth, allowing them to
efficiently and objectively sort maps from good to bad, check for cheating, and identify obvious gaps in quality
signifying grade boundaries. From our experience in 2020 and 2021, these maps are among the best products students
have produced in remote summer field alternative courses. In the course of mapping most students recognize that
nearly 100% of the valley walls have talus or debris flow deposits at their base; many recognize the variable density
of river terrace risers along the length of the valley.

## 185    2.4 Geologically-Reasoned Decision-Making in Yosemite Valley Exercise

Part IV of the module builds on the detailed geomorphic mapping the students did in Part III, which in turn builds on
the knowledge gained in Part I's VFT. In a KMZ file students are provided with two real world hazard lines that
Yosemite National Park uses for planning and preparedness: the known extent of the January 1997 flood in the valley,
approximating a 100-year flood, and a rockfall hazard line that considers talus slopes and isolated boulders but not
rock avalanches (Stock et al., 2014) (Figure 5). Students immediately recognize that some places in Yosemite Valley
are expected to be susceptible to both floods and rockfall, and that most of the valley floor is susceptible to one of the
hazards. Looking at the hazard lines overlain on the Google Earth satellite imagery, students can readily see what
existing park infrastructure is within or near the hazard extents.
Students are tasked with using both their geomorphic mapping and the hazard lines to (1) provide recommendations
for existing Yosemite Valley infrastructure that could be relocated to a less hazardous location, and (2) identify
locations that would be suitable for additional development with varying levels of risk (e.g, a hotel, visitor center,
storage facility, or parking lot). Students turn in a KMZ file indicating their recommendations and a technical report
written as if they were consulting for the park (an oral presentation would be an effective format too). It is emphasized
that the recommendations students provide must be well-reasoned and geologically-sound. Students should consider
the nature of the respective hazards (e.g., rockfalls occur instantly and without warning and can be fatal, floods in the
valley are predictable several days out and are rarely fatal) and the facility use (parking lots in flood zones can perhaps
be evacuated with limited damage, a seldom-visited storage yard is better near a rockfall hazard zone than a campsite,
etc.) when providing their recommendations. Students get particularly invested seeing how the numerous facilities are
placed throughout the valley. By the end of the exercise they gain an appreciation of the precious little real estate
available for further development in the valley, that natural hazards are an active part of Yosemite Valley, and that
there are considerable challenges associated with making decisions that affect the safety of over 4 million people a
year.



## 3. DISCUSSION

### 3.1 Designing Virtual Field Trips with Google Earth Web

The Projects feature of Google Earth Web is a robust and adaptable format for semi-immersive virtual field trips that can be created with relative ease and presented intuitively. Instructors and students alike can learn the basics of project creation in about 15 minutes, making it an efficient format for instructors and also suitable for students to create their own VFT as part of a course's final project. The VFT can be made readily available on web-connected tablets (via Google Earth) or computer (via web browser such as Google Chrome) using a simple web link. The abundant fair-use imagery available through the Google Earth Engine (topography-draped satellite, 360° street-level, user-submitted 360° photospheres) often means VFT creators do not have to start from scratch. Using a 360° camera or the Google Street View app the VFT creator can also upload their own photospheres. The ability to not only geotag a field trip stop as a point on a map, but to curate a precise starting view for that stop (e.g. zoom level, oblique 3D vantage, particular look direction in a photosphere) allows the creator to draw attention to the vantage most directly relevant to the learning goals for that stop. Text, images, and videos can be added to a sidebar supporting the stop's view; annotated photos of a similar or identical view can be particularly illustrative. However, each stop's view is not fixed, allowing the viewer to explore "off-script", which can improve their situational awareness and opportunities for independent learning. Advanced creators can customize the html for a given html stop, providing an opportunity to add custom icons, narration, and built-in quizzes. Though less self-contained, hyperlinks offer an opportunity to send students to external web addresses for background information (e.g. Wikipedia), quizzes or surveys (e.g. Google Forms), and even web-hosted 3D models of outcrops or hand samples (e.g. Sketchfab).

As a result of the COVID-19 pandemic, Google Earth Web VFTs were created for several locations instructors would otherwise be taking students on physical field trips, such as Greece (Evelpidou et al., 2021b) and Colorado (Mahan et al., 2021). A notable pre-pandemic general public-focused implementation of Google Earth Web Earth projects as VFTs is "Streetcar 2 Subduction" hosted by the American Geophysical Union (Rowe et al., 2020; https://www.agu.org/streetcar2subduction). On their dedicated web page eight separate Earth projects are linked that cover geological field trips in the San Francisco Bay Area, building on the classic field trip guidebook "A Streetcar to Subduction and Other Plate Tectonic Trips by Public Transport in San Francisco" by Wahrhaftig (1984). While these trips are fully functional as VFTs, they include information on transportation and safety logistics, and are designed as self-guided walking tours. Because cellular data reception is generally excellent in these urban-adjacent field trip areas, a self-guided participant can use their GPS position (as a blue dot) to navigate from one stop to the next within the VFT frame, and access the text, annotated images, and videos provided at each stop along the way. There is a vast collection of existing field trip guidebooks that could be adapted into immersive VFTs to reach a broader audience. Despite a global pandemic, United States national parks still hosted 237 million visitors in 2020; the top ten visited parks all prominently feature geology (NPS Press Release, 25 February 2021). One could imagine the value of linking immersive VFTs on park websites to help visitors plan their physical trips and phone-scannable QR codes displayed outside visitor centers that would allow self-guided trips.



Currently there are several limitations of the Google Earth Web project format that are worth discussion. There is no
functionality to add georeferenced layers to a Google Earth Web project, such as a geologic map that could be turned
on or off over a landscape or a folder of data points that are distinct from field trip stops. Analysis is limited to
measuring distances and areas. Google Earth Web also currently does not have the ability to switch between different
imagery dates, a stellar feature on Google Earth Pro (desktop) that better highlights landscape changes (e.g. before
and after a wildfire). If there was a more straightforward way to cache all imagery and media related to a project, the
VFT could be taken to remote field locations and actually be used as more of an interactive field trip guide. At the
moment there does not seem to be a way to directly embed 3D models into Google Earth Web project sidebars; we
attempted to embed web-hosted Sketchfab, SketchUp, and 3D PDF models but were blocked by script or cookie
permissions. The ability to add an embedded 3D scan of a rock sample or a detailed outcrop model would likely add
to a VFT viewer's experience (these can still be hyperlinked though). Adapting the Google Earth Web project format
to Google Earth VR would certainly boost the immersiveness of a VFT, but at the cost of being a less accessible
format for many due to the specialized equipment currently needed for VR (e.g. Hagge, 2021). For public outreach
efforts and student engagement it would also be beneficial if statistics could be accessed about the number of viewers
and how long they viewed the VFT.
**3.2 Comparison to Other VFT Formats**
ArcGIS Online allows multiple layers of vector data to be viewed on a single customizable map interface (though
currently custom raster layers are not supported) and provides the ability to analyze and filter the data (e.g. West and
Horswell, 2018); however,this format does not really support a presented or guided structure and so additional
resources are needed to support a VFT. ArcGIS StoryMaps, best characterized as a map-centric dynamic webpage, is
another adaptable format suitable for geographically oriented tours or narratives (for VFT examples see Evelpidou et
al., 2021a; Senger et al., 2021); StoryMaps offers a more structured and less immersive VFT option than Google Earth
Web projects. Other more specialized formats exist on pay-to-create software platforms. StoryMaps GPS (e.g.
California State University Fullerton's Yosemite Fire & Ice tour accessed at
https://www.travelstorys.com/tours/154/Yosemite%20National%20Park) offers a similar presented format as Google
Earth Web with an overview map and field stops that can be selected from a table of contents, but lacks the key ability
of associating a discrete view with each stop.
Arizona State University hosts many publicly accessible VFTs built in the SmartSparrow software platform (Mead et
al., 2019; https://www.vft.asu.edu). These trips are largely photosphere-centric with built-in links to video and image
pop-ups, and links to additional photosphere stops; the result is an immersive experience with high production value
(there are even ambient bird sounds) in a completely self-contained format. Several educators have been
experimenting with using videogame platforms (e.g. Minecraft, itch.io) to allow exploration-based field simulations
in both scanned real world sites and fictional environments (Needle et al., 2021; Rader et al., 2021); these formats
typically require more commitment on the part of the educators to design and implement but can be engaging, allow
interaction between students, and mimic the freedom of mapping a region for the first time. Virtual Reality (VR)
experiences offer the most immersive VFT possibilities by allowing the viewer to have the virtual environment





surround them (e.g. Peterson et al., 2020; Hagge, 2021; Métois et al., 2021); unfortunately, the specialist VR goggles,
software, and computer are likely unavailable to most students at home and students will almost certainly need to take
turns in a classroom setting. Arguably one of the more flexible formats for presenting VFTs remains a custom html
webpage; this format allows embedding of maps (e.g. ArcGIS Online, Google My Maps) and 3D outcrop or rock
sample models (e.g. Sketchfab), as well as external web links or links to download supporting materials (e.g. Bond
and Cawood, 2021).

## 3.3 Building Effective Learning Through VFTs

Undoubtedly a major advantage of VFTs is their immediate accessibility to locations around the globe and beyond.
Additionally, VFTs through Google Earth provides opportunities for students to participate in cognitively engaging
and interactive learning experiences, which have been found to improve student outcomes in STEM courses (Freeman
et al., 2014; National Research Council, 2011). Freeman and colleagues (2014) synthesized the findings of 225 studies
that reported data on examination scores or failure rates when comparing didactic instruction versus active learning
on student performance in undergraduate STEM courses. Their results revealed that learning outcomes were
significantly better for students whose experiences included active and interactive pedagogical practices.
VFTs using Google Earth allow for students to engage in guided discovery learning (Mayer, 2004). Virtual learning
environments based solely on discovery learning allow students to independently explore and solve problems with
little to no guidance (Lee and Anderson, 2013; Mayer, 2004). However, one drawback to this type of learning
environment is that it can lead to excessive cognitive demands, especially when it comes to users with limited domain-
specific expertise. The embedded instructional prompts that the Google Earth VFT format affords enables the
instructor to promote guided discovery of the environment as it allows for the integration of direct instruction into
discovery learning (e.g., Lee and Anderson, 2013; Mayer, 2004), providing students with the scaffolding necessary to
navigate and learn from such a perceptually and informationally rich environment.
Google Earth-based VFTs are also an effective means to scaffold students' spatial skills. Spatial skills are a set of
cognitive skills that enable us to manipulate, organize, reason about, and make sense of spatial relationships in real
and imagined spaces (e.g. Atit et al., 2020; Newcombe and Shipley, 2015; Uttal et al., 2013). Field geology heavily
relies on the use of spatial skills as the goal is to use present day spatial properties to infer the geologic history of a
region (e.g., Atit et al., 2020; Shipley and Tikoff, 2016). In particular, identifying the relevant spatial properties in the
field requires the geologist to focus their attention on the critical spatial information (e.g, the orientation of the rocks
on a bedding plane). Focusing attention on the important spatial information involves actively ignoring many other
aspects of the scene (e.g., the orientation of the tree on the outcrop or the size of the minerals - geologic features not
pertinent to the problem at hand). The spatial skills used to identify relevant information for further cognitive
processing is called disembedding in the geosciences (Manduca and Kastens, 2012; Reynolds, 2012) and selective
attention in psychology (Moran and Desimone, 1985). Novices find disembedding to be difficult (Coyan et al., 2010;
Shipley and Tikoff, 2016). Google Earth allows the user to remove the extraneous irrelevant information (e.g.
vegetation) from the scene, bolstering geologically-relevant disembedding tasks for novice users.



### 3.4 From General Public VFT to Geologically-Reasoned Decisions

Through our Geology of Yosemite Valley module, we provide an example of how a VFT can be designed to be approachable to a broad general public audience and at the same time serve as background information for an upper-division student mapping project. Hyperlinking technical words to external resources is an invaluable way of unobtrusively broadening the target audience. Distilling VFT stops to the most critical learning goals (especially with annotated images and videos) and encouraging interaction with the immersive view at the stop likely increases learning and engagement. Designing exercises that require students to utilize a combination of data they create (e.g. mapping) and real world data (e.g. hazard data, stream gauge data) to justify decisions trains them to develop professional skills and increases their investment in the task. The progression from VFT to mapping to professional-style consultant report produced the highest quality work of any of the eight modules and exercises covered in UCR's 2020 and 2021 virtual Summer Field Geology course offerings. Anonymous student feedback from the 2021 offering indicates that the Geology of Yosemite Valley module garnered the most positive response. 50% of the class said the module was their favorite component of the course and 60% said they learned the most from it; no students thought it was their least favorite or that they learned the least from it (unfortunately no anonymous feedback was collected from students in 2020).

### 4. CONCLUSIONS

The Projects feature of Google Earth Web is a robust and adaptable format to create rich and engaging virtual field trips with relative ease. The abundant fair-use imagery built into the Google Earth Engine (satellite, 360° street-level, user-submitted 360° photospheres) allows for immersive stops that enable creators to point to specific features, but also encourages viewers to learn by exploration, mimicking an advantage of in-person field trips. Many media types can be directly embedded (images, videos, narration) or hyperlinked (websites, 3D models, quizzes, etc.) to customize the VFT presentation and adapt to a broad range of knowledge levels ranging from general public to upper-division major students. Where cellular coverage exists, this VFT format can also be used for self-guided field trips. When properly implemented, a general public-oriented VFT can be used as background for mapping exercises, which in turn can be used to encourage students to support geologically-reasoned decision-making.

### ACKNOWLEDGEMENTS

The authors wish to acknowledge Allen Glazner for critical review of the VFT's scientific content. Rachel Atkins, Ryan Petterson, Alberto Pizzi, Nadia Salvatore, and Ben van der Pluijm are thanked for peer review that improved the VFT's functionality. Thank you to Christopher Atchison, Anne Egger, Basil Tikoff, Kurt Burmeister, and Katherine Ryker for spearheading the joint NAGT-IAGD Designing Remote Field Experiences effort. Thanks also to participants in the Virtual Class-Related Field Trips working group.

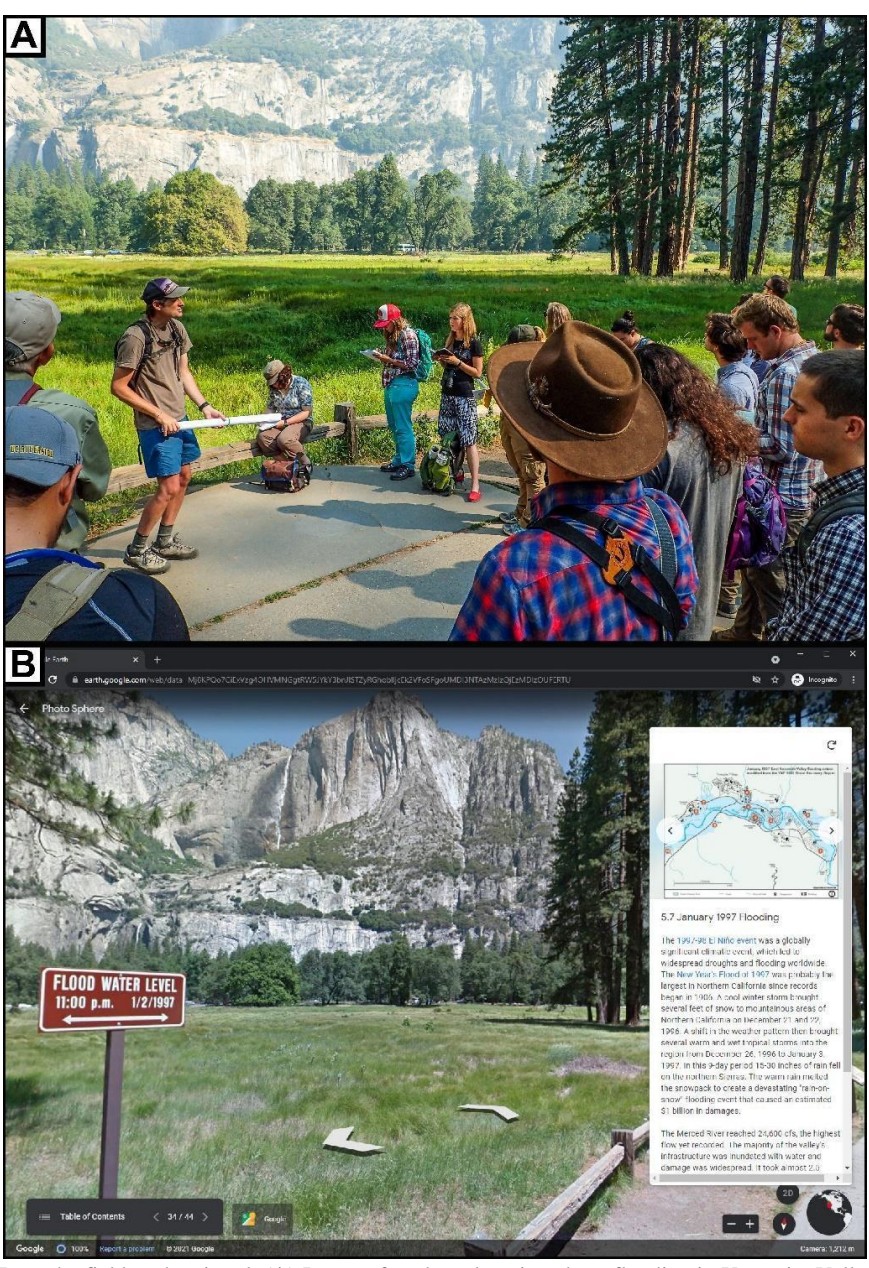

**Figure 1.** From the field to the virtual. (**A**) Image of students learning about flooding in Yosemite Valley as part of an in-person field trip in 2017. Just to the left of the view is a ~1.5m tall sign marking the peak flood water level on January 2$^{nd}$ 1997 (seen in B), a striking location to discuss flood hazards as students look over the meadows, trails, and roads that would have been inundated. (**B**) A corresponding stop in the Geology of Yosemite Valley virtual field trip that utilizes a precisely chosen © Google Street View look direction and zoom level to provide the same discussion of 1997 flooding. Students can pan the view to get a similar sense of flood inundation around them. Text and hyperlinks are in the sidebar. A historic photo of the flooding event, map of the flooding extent, and a NPS-produced video on Yosemite rain-on-snow flooding fill the full-screen when clicked on. Virtual field trip is an Earth project created with © Google Earth Web viewed in © Google Chrome web browser.





**Figure 2.** Creating a VFT Earth project in © Google Earth Web. (**A**) A view highlighting the interface for editing a project (i.e. VFT). All changes are instantly saved to the cloud (the creator's Google Drive). (**B**) A view highlighting the interface for editing a feature (i.e. virtual field trip stop). Many customizable options exist.



**Figure 3.** VFT stops highlighting learning through guided exploration. Left panels show the VFT stop view in ©
Google Earth Web; right panels show the corresponding annotated image embedded in the stop's description. VFT
stops showcase (**A**) the medial moraine at the junction of Tenaya Creek and Merced River, (**B**) exfoliation-related
rockfall sources of different ages on the slopes of Half Dome, and (**C**) regional joints exposed in the cliff faces of
Sentinel Rock. By simulating the dynamic view in a figure, the viewer can better visualize the key features
emphasized. In each instance the dynamic view encourages the viewer to zoom in closer to a feature or search the
surrounding area for similar features.



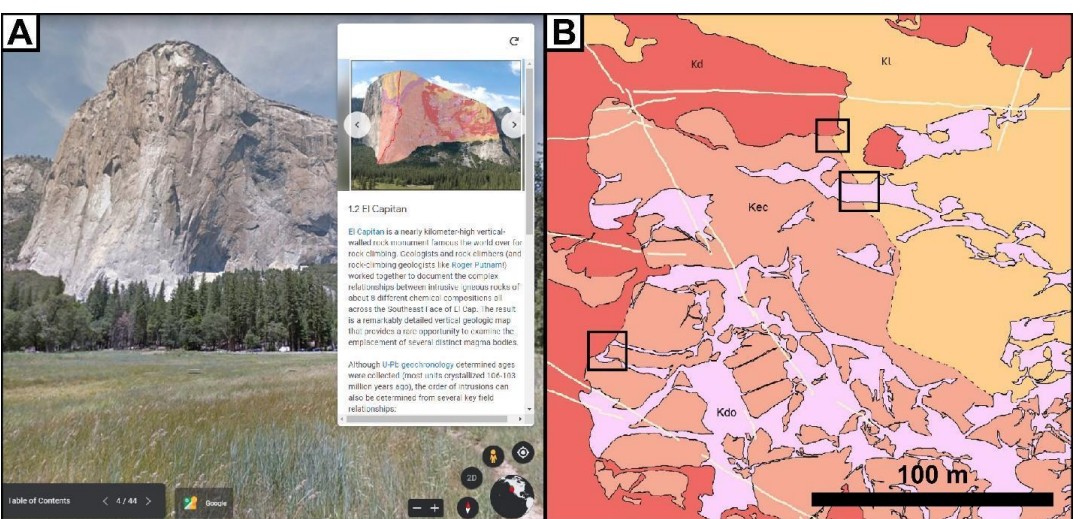

**Figure 4.** Introducing a complex relative dating exercise. (**A**) Stop on the Geology of Yosemite Valley VFT (© Google Earth Web) highlighting the vertical geologic map of El Capitan created by Putnam et al. (2014) and providing an overview of the knowledge (cross-cutting relationships, law of included fragments, gradational contacts) necessary to relatively date intrusive units. (**B**) An excerpt from the modified Putnam et al. (2014) El Capitan geologic map that students use to determine the relative timing of eight intrusive units. Black boxes highlight three areas that provide unambiguous relationships between two or more units.

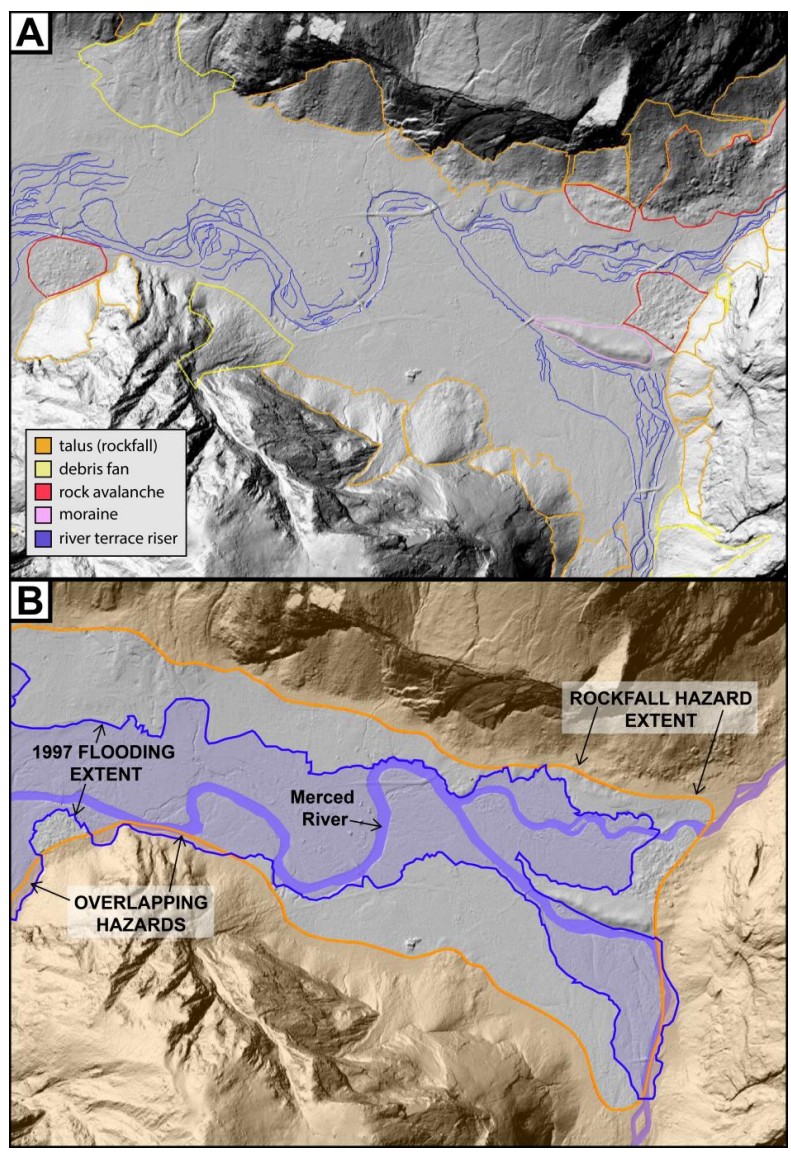

**Figure 5.** From geomorphic mapping to hazard planning. (**A**) An excerpt from a virtual summer field student
geomorphic map graded "highly competent" (still room for improvement) submitted as a Google Earth KMZ file
(final product of Part III of the module). Basemap shown in both panels is a hillshade of bare-earth airborne lidar data
collected by NCALM (2006), web-served by OpenTopography, and funded by NSF (publicly accessible). Students
use a combination of the lidar visualizations and satellite imagery provided to map deposits based on texture, slope,
and valley position. (**B**) A visualization of two hazard lines utilized by Yosemite National Park and provided to
students. Blue line indicates the maximum extent of January 1997 flooding (approximating a 100-year flood). Orange
line indicates the expected extent of rockfall hazard within the valley (i.e. beneath the valley walls) from Stock et al.
(2014). In Part IV of the module students use their geomorphic mapping and these hazard lines to examine the hazards
posed to existing infrastructure and identify areas of lower hazard suitable for further development. Geologically-
based justifications are expected for all of their recommendations. Location, scale, and orientation are deliberately
excluded from this figure. Width of valley floor here is ~1 km.



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
