# Peer review of "From Virtual Field Trip to Geologically-Reasoned Decisions in Yosemite Valley"

_Geoscience Communication, 2021_

## Author Comment (AC1)

**Review 1 Citation**: https://doi.org/10.5194/gc-2021-28-RC1

Review of "From Virtual Field Trip to Geologically-Reasoned Decisions in Yosemite Valley" by Barth et al
Dear NC Barth and co-authors,

The present paper is interesting and well-written.  It presents an elegant Virtual Field Trip (VFT), the authors' reasoning behind this VFT, some analysis on the technical solution for this VFT, some thinking on VFTs in general, and the instructor's impression of the learning outcomes of the students exposed to this VFT. The VFT itself looks polished and is easy both to understand and to use, and will likely be interesting to many. I have only a few comments, and I think this paper only needs very minor revisions to make it accessible to as many as possible. Many parts are somewhat descriptive and describe basic functionality in the technological solution, or are somewhat repetitive. This could be streamlined a bit.

Thank you for your detailed review, and in particular, your efforts to make our study more broadly accessible. We are pleased you found our manuscript needing "very minor revisions". We address your specific comments as individual responses below and have incorporated the majority of your suggestions.

My specific comments to the paper can be found below.

Sincerely,

Christian Haug Eide

University of Bergen

October 5th, 2021

Main points 1: Present link to virtual field trip earlier in paper

Right now, the link to the VFT is not presented before Section 2, Line 55. I believe many readers would like to see the VFT itself as early as possible, and I think it would be useful to place this link in the first paragraph of the introduction.

We have added a link to the four-part activity in the Introduction and have added a link to the VFT in the Figure 1 caption (first called out in the Introduction). Figure 1 more directly relates to the VFT component of our activity than the Introduction does generally. We think including the link in the first paragraph of the Introduction would disrupt our narrative flow and is more suitable at the end of the Introduction.

2: US-centric jargon

Many terms relating for example to how far students are into their study programme are US-specific and not immediately obvious to readers from other locations. Examples include upper division, summer field, course, module, gradebook, major.

Yes, we recognize now this is a bigger problem than we anticipated so thanks for helping us communicate effectively to a global audience. We removed all instances of "upper-division" and replaced with "advanced undergraduate" or "third or fourth year undergraduate" or "capstone" depending on the use. Reference to "grade-level" was changed to "knowledge-level". Reference to "grade" was changed to "score". Reference to "gradebook" was changed to "records of scores". The first instance of "courses" was changed to "educational experiences" with a description of US terms after. The new first instance of "course" in the manuscript is now followed by "(i.e. "module" in the European use)". To avoid confusion, we have removed our usage of "module" altogether and replaced it with "activity" throughout the text. "Summer field" now occurs in a parenthetical clause linking it to "capstone field geology educational experiences". "Major" is no longer used to describe a concentration in study.

3: Unnecessary sections?

I am not sure section 3.3 adds very much? (This might just be because I am a philistine). 3.4 also seems to me to be much longer than necessary?

Concerning Section 3.3: The focus of much of the manuscript is on the remote learning activity we developed, how the format of VFT we chose fits into it, and comparative advantages/disadvantages of our general approach. Section 3.3 provides heavily referenced discussion from state-of-the-art literature surrounding geoscience education- studies on how humans learn and what strategies have proven effective in learning. This discussion section seeks to expand our study's contributions by placing the advantages of our chosen VFT format into a broader context of learning and hopefully to encourage other VFT creators to consult documented strategies for effective learning. The other two coauthors consider these discussion contributions by our educational research colleague to be valuable and entirely appropriate to the call of the special issue to which we have submitted. We have actually decided to lean into this section more (partly in response to more directed comments below) by adding five references and sentences on cognitive load. Notably we added "Cognitive load refers to the load that performing a particular task imposes on the cognitive system (e.g. Paas and Van Merrienboer, 1994; Sweller, 2011). The amount of information one's cognitive system can process at a given moment is limited. Thus, the presentation of too much information, some of which is unnecessary information when it comes to solving the task, can result in artificially increasing the cognitive resources needed to process the relevant content. This is referred to as extraneous cognitive load by Sweller (2010), and it results in decreasing the efficiency and efficacy of the learner's cognitive system (see review of cognitive load theory by Paas et al., 2010)."

Concerning Section 3.4: This is by far the shortest section of four in the Discussion. We find every sentence to be on topic with the section header "From General Public to VFT to Geologically-Reasoned Decisions". We feel this section, largely aligned with the overall title of the manuscript, sums some of our lessons learned and logically leads the reader into the Conclusions section. We could remove the detailed course statistics, but note that elsewhere the reviewer asked for more of these if available, as did the other reviewer (which we did). We do not see a need to shorten this section and prefer not to remove it altogether.

4: Hillshade

It is not clear how the hillshade functionality was delivered in google earth? On lines 312-3 it is stated that Google Earth can remove vegetation, but this is not standard functionality as far as I know. This should be described better.

We did not mention hillshade or intend to suggest that Google Earth digitally removes vegetation, but we do see the potential for confusion here. We removed the "e.g." concerning vegetation entirely here; we think the sentence at line 308 adequately provides two examples of extraneous irrelevant information to the task at hand.

Minor points:

L17, L299 and 301: The term scaffolding is used but it seems vague to me what this actually means. Perhaps the authors could clarify this?

Literally, scaffolding is a network of platforms temporarily applied to allow workers to access and construct buildings. The term is pervasive in educational literature since the 1970s. Generally scaffolding is where a teacher provides a lot of support initially to aid students and then progressively removes that support as the student gains confidence and ability.

For the reviewer's sake, here is a useful summary of scaffolding as a method in education: https://www.gcu.edu/blog/teaching-school-administration/what-scaffolding-education

For the sake of readers less familiar with pedagogical terms, we have added a parenthetical after the first in-text appearance that clarifies it as "additional learning supports that can eventually be decreased with increasing ability" and also added a reference to provide further background.

L30: "Most instructors" is a bit vague.

We do not have exact statistics available to us unfortunately. The corresponding author knows of three out of at least fifty schools (6%) that were able to run in-person field activities largely as they would have pre-pandemic. It is perceived that these are a vocal minority and the actual percentage is likely to be lower. It is beyond the scope of our study to poll all of these courses. While vague, we prefer leaving our descriptions as "a very small minority" paired with "most instructors" as we have. No change made.

L32: "began organizing and meeting virtually in March 2020" – what was this forum called?

To the best of our knowledge it started as "Designing Remote Geology Field Courses" and eventually became "Designing Remote Field Experiences". We have added the latter forum name to the text at this location for clarity.

L44-45: Especially here, it should be clarified what a module is and what a course is.

The first author knew of many conflicting word usages in education between countries but was ignorant to the particularly troublesome uses of "course" and "module" so thanks for the enlightenment. The first instance of "course" (before this location) now has a parenthetical clause indicating equivalence to "module" in the typical European use. To avoid confusion, we have removed our usage of "module" altogether and replaced it with "activity" throughout the text.

L83: "What makes placemarks truly standout" – unnecessarily loaded, change wording.

Ok, sure. This text has been changed to "A key advantage of placemarks…".

L100-106: This is perhaps a bit too long and detailed?

At this suggestion we removed two supporting sentences here, shortening the length from 7 lines to 4.5 lines.

L122: Exceptional quality. It would be better to provide some objective measures of quality such as resolution, or "largest feature that can be recognized"

At this suggestion we have added supporting text here mentioning that individual trees and boulders can be resolved.

L133, 134: Loaded terms good, excellent. Text should be rewritten here to be more formal and more descriptive.

Text has been rewritten to be more formal. "Good" has been removed; "effective" has been substituted for "excellent." We do not think more description is necessary for this sentence.

L145: "steeped in the history of rock climbing". Vague, should be presented in a more informative manner.

We point to the Oxford English dictionary definition of "steep" provided via Google dictionary: "steep (verb) (1) soak (food or tea) in water or other liquid so as to extract its flavor or to soften it. "the chilies are steeped in olive oil" (2) surround or fill with a quality or influence. "a city steeped in history"." Our use of steep here is well aligned with the second usage of the word; we argue our use is not vague and is entirely informative. We see no concern over this word choice and prefer not to change it.

L151: "An instructor-only file is provided with verified credentials." It is unclear to me what this means.

We have changed the text to clarify that a file with solutions to the exercise is available through the SERC website with verified instructor credentials.

L178-9: "are held to a professional standard." It is unclear to me what this means.

We have rewritten to "the quality of the work is expected to be comparable to that produced by an entry-level professional geoscientist".

L181-2: " these maps are among the best products students have produced in remote summer field alternative courses". Compared to what? Judged on what metrics? What is so good about them and how are these better than the ones they made before/in other modules?

We have added "UCR's" before "remote summer field alternative courses", which hopefully clarifies the many questions here and further emphasizes that this is based on our experience teaching at UCR in 2020 and 2021. We do not wish to get as granular as discussing student scores and rates of success in

meeting the many learning objectives across multiple projects, especially given the relatively small sample size (n=22). The following sentence provides two tangible examples of students meeting learning objectives; other mapping projects did not exhibit the same degree of completeness, accuracy, or understanding.

L257: Another point that should be pointed out, I suppose, is that there might be no guarantee for longevity for these VFT products. I suppose google could change their solutions and all this becomes obsolete? Or are there guarantees against that?

Google Earth is one of the most robust, long-lived, and ubiquitous pieces of software the authors are familiar with (largely unchanged over 20 years) and Google continues to support it in a reverse compatible sense to ensure no data loss. But yes, as with any format, longevity is not guaranteed and we have added a sentence to this discussion to acknowledge this.

L296: What exactly is meant by "excessive cognitive demands" here? I can think of very many experiences in higher education that places much higher cognitive demands on students than such a VFT. Also, what is meant by domain specific expertise here (i.e. which domain are you thinking about – geology, general computer skills, google earth expertise)?

What we mean by excessive cognitive demands is extraneous cognitive load. We have revised the paragraph to be clear as to what we mean by extraneous cognitive load and how the Google Earth VFT could potentially decrease students' cognitive load, facilitating learning. Additionally, we have clarified that what we mean by domain-specific expertise is knowledge about geology and about conducting fieldwork.

The relevant text is now: "Cognitive load refers to the load that performing a particular task imposes on the cognitive system (e.g. Paas and Van Merrienboer, 1994; Sweller, 2011). The amount of information one's cognitive system can process at a given moment is limited. Thus, the presentation of too much information, some of which is unnecessary information when it comes to solving the task, can result in artificially increasing the cognitive resources needed to process the relevant content. This is referred to as extraneous cognitive load by Sweller (2010), and it results in decreasing the efficiency and efficacy of the learner's cognitive system (see review of cognitive load theory by Paas et al., 2010). With the aim of reducing the learner's cognitive load, the embedded instructional prompts that the Google Earth VFT format affords enables the instructor to promote guided discovery of the environment by allowing for the integration of direct instruction into discovery learning (e.g. Lee and Anderson, 2013; Mayer, 2004). This direct instruction provides students with the scaffolding (i.e. additional learning supports that can eventually be decreased with increasing ability) necessary to navigate and learn from such a perceptually and informationally rich environment (e.g. Lee and Dalgarno, 2011)."

On behalf of the authors, Nicolas Barth (nic.barth@ucr.edu) 28 October 2021

END

---

## Author Comment (AC2)

**Review 2 Citation**: https://doi.org/10.5194/gc-2021-28-RC2

Review of: From Virtual Field Trip to Geologically-Reasoned Decisions in Yosemite Valley.

Nicolas C. Barth, Greg M. Stock, and Kinnari Atit

Dear Simon Buckley,

This valuable contribution provides a concise outline of what appears to be a well-constructed virtual field trip (VFT) within Google Earth that can be used for entry to high level geoscientists. It offers some reasoning of the VFT, an overview of the components, and a discussion on the technicalities of using Google Earth as a platform. In addition to comparisons with some other forms of VFTs, and insights into how VFTs can provide an effective learning platform. My comments are listed below, and I recommend minor revisions.

Sincerely,

Jessica Helen Pugsley

University of Aberdeen

17/10/21

Thank you for your detailed review; we found many of your comments improved our manuscript. We are pleased you found our manuscript needing "minor revisions." We apologize that it seems like at least two key links were broken (beyond our control) in the time between our submission and your review. We address your specific comments as individual responses below.

Main Points:

1: VFT Link

A direct link to "four-part Geology of Yosemite Valley module" would benefit from being added to the introduction if it is publicly available as stated. In addition to the general "Teaching with Online Field Experiences" link in section 2.

A direct link to the four-part module (now called "activity") was provided in the second line of text after the Introduction (Line 55), but this was broken in the time between submission and review and redirected to the general "Teaching with Online Field Experiences"- we apologize, this link has now been updated. We were hesitant to include the awkwardly long direct link to the VFT (https://earth.google.com/web/@37.73678425,-119.58078081,1418.42744788a,93163.53588265d,35y,0h,0t,0r/data=MicKJQojCiExVzg4OHVMNGgtRW5JYkY3bnJlSTZyRGhoblljcEk2VFo6AwoBMA?authuser=0) but have used a URL shortener so that we could include a less unwieldy link at Line 64 (https://bit.ly/2Zbn3R7). We now include the link to the overall activity towards the end of the Introduction, and also include the VFT link in the Figure 1 caption (called out in the Introduction) as it is more on topic there than in the Introduction.

2: Geology vs Geomorphology

In places it is unclear if the students are interpreting geological features or geomorphological features. If students are describing geomorphological features but relating them to the geology of the area, this should be clarified potentially with examples, or where appropriate, the term geomorphology used.

We searched through our manuscript with this comment in mind and cannot find any locations where we find the descriptions unclear or needing additional examples. Generally, we view geomorphology as a sub-discipline of geology and most components of the project do extend beyond geomorphology. Part I (the VFT) encompasses tectonics, igneous rocks, bedrock joints in addition to geomorphology so is best described by its title, Geology of Yosemite Valley VFT. Part II does not involve geomorphology at all and is a purely "geological" exercise. Part III is a geomorphic mapping exercise; we believe we make this clear and note that "geology" does not appear in its description. This reviewer comment seems to be most directed at Part IV being classified as a geological exercise instead of a narrower-in-scope geomorphic exercise. Please see our detailed response below regarding Line 185 where we attempt to articulate our choice in using "geology" vs "geomorphology."

3: Introduction

The introduction is clear and concise however it lacks references and context, perhaps sections of the discussion (3.2) outlining VFT Formats, would be better placed here.

Thank you for this comment. We did consider this suggestion, but mostly prefer the current organization. The manuscript title and much of the content of the manuscript extends beyond the VFT component of our activity (one part of four) and we do not wish to overburden the Introduction by diving into the specifics of different VFT formats. We instead prefer to use the Introduction to identify the pandemic-driven need to create new virtual field materials and to set the stage for introducing our four-part activity which was created in response. We think a Discussion section is the appropriate place to weigh the pros and cons of different VFT formats. We have added four references in support of the general conditions and challenges educators and students faced during the pandemic (Arthurs, 2021; Phillips et al., 2021; Walker, 2021; Rotzien et al. 2021). We acknowledge this is still a low number of references here, but believe this helps us obtain a clear and concise Introduction, and do not think we are excluding a needed reference. We think that our use of references in the manuscript is otherwise thorough.

Minor points:

L14-15: "Upper-division" unclear within this context external to US to my knowledge, consider another.

We have changed the usage here to "advanced undergraduate students," which we think should be more broadly understood.

L17: 'Scaffolds' as a term in this context not particularly clear, consider another word or phrase.

"Scaffolding" has a particular meaning in geoscience education literature and this is what we intended to use here. However, because this is the Abstract and we do not want to burden it with jargon, we agree with a different word choice here, although it somewhat diminishes the intended meaning. We have changed the use here to "supports".

L31: Consider adding some references to this, there are several examples of such alternative courses now published. More generally there is not a single reference within the introduction, while much of what is mentioned is either general knowledge or specific description of the Yosemite Valley VFT, there are sentences which could be substantiated through references.

As stated above, we chose to use the Introduction to provide context to our newly created activity (the pandemic, the immediate need to assemble and create remote materials, and our typical class format at UCR), which allows us to get into the activity as efficiently as possible. None of the topics covered in the Introduction have obvious necessary needs for references and so we tend towards a more streamlined and efficient Introduction. Nevertheless, we have added four references that highlight general challenges concerning field geology education in the time of COVID-19 (Arthurs, 2021; Phillips et al., 2021; Walker, 2021; Rotzien et al. 2021). Overall we would consider our study to be heavily referenced in comparison to the similar studies we are referencing and that have been submitted to this special issue.

L43-45: 'instructors thought' vague, do you have any data to support?

Yes, this is based on student assessment results. We do not intend to get so granular as to show students' grades and different rubrics are used, which is why we initially preferred to point to the instructor's perception instead. To be less vague we have changed the text here to clarify this is based off of student assessment results.

L43-44: "course" and "module" both used, consider clarifying.

The first author knew of many conflicting word usages in education between countries but was ignorant to the particularly troublesome uses of "course" and "module" so thanks for the enlightenment. The first instance of "course" (before this location) now has a parenthetical clause indicating equivalence to "module" in the typical European use. To avoid confusion, we have removed our usage of "module" altogether and replaced it with "activity" throughout the text.

L46: "potential" used twice, consider another word or phrase.

Good catch. We have changed the first "potential" and in support of your previous comment removed an instance of the word "module".

L55: this link is to a general Teaching with Online Field Experiences page and not the VFT mentioned, is there a direct link to the VFT if publicly available?

Between submission of our manuscript and your review, we did notice SERC restructured their links to the collection of teaching activities including ours. The link has been updated and we certainly hope does not undergo another modification beyond our control. We apologize if you were unable to access the activity webpage during your review. It can be found here: https://serc.carleton.edu/NAGTWorkshops/online_field/activities/237092.html

L57: Does the module work on a smart tablet within google chrome and Google Earth Pro? If so, add 'touch-screen' or 'remove trackpad or mouse'.

"Full functionality" is the key phrase we use here. The mapping activity comprising Part III really needs a desktop version of Google Earth Pro to complete. The Part I VFT works great on smart tablets. We do not wish to burden the reader with additional text explaining what works and does not for each exercise here. The sections that describe each activity component in detail do provide information that should suitably answer your question. See Lines 101-103 and 165-166.

L72: what is described are commonly referred to as 'live edits' or a 'real-time' project/document.

We intended to avoid introducing more jargon than necessary since this could fall out of vogue (hence our explanatory description which happily you were able to understand), but to clarify for our current audience we have added a parenthetical identifying this as "real-time editing".

L84: 'look direction' vague, consider 'view orientation'.

"Look direction" is commonly used to describe camera positions and satellite view orientations; we thus would not consider this vague but nevertheless have changed the text to "view orientation". For consistency we have also changed the two other instances where we had used the term "look direction".

L84: Clarify what is meant by 'imagery'?

The next three sentences (Lines 84-89) does exactly that. Clarifying here does not improve understanding. No change made.

L103: unclear what "probably ideal in most situations" means

The other reviewer also suggested modifications to this sentence. We removed "and probably ideal in most situations" and will leave it to the reader to make this call.

L122: The "exceptional quality" of Google Earths photogrammetry mentioned is through the google earth '3D Buildings' layer, which covers numerous cities globally in addition to some national parks (mainly within the US). The raw data is collected through low-flying arial photogrammetry, not satellite imagery as this contribution describes. This data is currently restricted to cities and national parks, consider mentioning. If you turn off this layer you will see the quality of the satellite imagery below area in figure 3c.

Thanks for this comment. Yes, you are absolutely right that the referenced imagery is a product of low-flying aircraft and the availability of the 3D Buildings layer at this location, not satellite-based imagery. We have modified our explanation here to include mention of the "3D Buildings" layer (though this will inevitably be renamed as more natural places gain coverage) and changed our explanation of this layer to "photo-textured 3D models derived from low-flying aerial photogrammetry". We added "current" and "other popular outdoor recreation areas" to further explain coverage. As an FYI to the reviewer it is not correct that current coverage is limited to cities and national parks- it is more random than that. Going to earth.google.com, clicking on "Voyager" on left side-bar, then "Layers", then "3D Imagery in Google Earth" shows helpful outlines of current extent. If we suspected that this link would remain unchanged (we do not), we would share it in our manuscript so that others could check whether their areas of interest had coverage.

L122: The use of "exceptional, excellent, and good" noted here and elsewhere in paper, try to use more objective language.

The other reviewer agrees with you here. After our use of "exceptional" here we clarify that the resolution is such that individual boulders and trees are resolvable. This is the only instance of "exceptional". Our only instance of "excellent" is in the sentence "Because cellular data reception is generally excellent in these urban-adjacent field trip areas, a self-guided participant can use their GPS position (as a blue dot) to navigate from one stop to the next within the VFT frame, and access the text, annotated images, and videos provided at each stop along the way." We have remove "excellent" and instead change the sentence to "Because there is often cellular data reception in these urban-adjacent field trip areas, …". There is one instance of "good", "objectively sort maps from good to bad…". We have removed "good" by changing this to "objectively sort maps based on quality…"

L137: "It is a simple matter" unnecessary, consider "to continue".

We agree. The text has been changed here.

L172-173: Where is the lidar data sourced from that the students use to make hillshades? If data external reference source. Also clarify what a hillshades map is.

We have defined a hillshade ("greyscale NW-illuminate representation of surface"), reworded to clarify that students are provided hillshades to interpret (not create), and provided a doi link to the raw dataset (https://doi.org/10.5069/G9GQ6VP3).

L176: Consider removing "fairly".

Removed.

L178: "Summer field" vague, not a term commonly used in Europe.

We have changed this to "capstone field geology" to be more globally applicable.

L181: How did you measure the quality; do you have the data?

We are not completely sure what is meant by these questions but attempt to answer here. The maps are assessed based on a rubric that includes scores for correctnesss, completeness, and neatness as mentioned a few lines above. We have also added a sentence clarifying that an instructor-produced map is available for download with verified instructor credentials to further guide the assessment of student mapping "quality".

L182-184: Is there data of how many students recognise these features, rather than stating 'many' or "nearly 100%".

We now present the data as percentages and total numbers of students. For reference this is 12 students in 2020 and 10 students in 2021. We consider this is a small dataset playing a minor role in our study; considering the many external factors students faced in 2020 and 2021 (apart from our teaching ability), we do not see any need to present this in a more detailed fashion, for example breaking out

statistics to compare between years.  "nearly 100%" was in reference to valley wall coverage, not an attempt to approximate statistical data; we have changed this to "nearly all". The text is now: "From our experience in 2020 and 2021, these maps are among the best products students have produced in UCR's remote summer field alternative courses. In the process of mapping, most students (82%; 18 out of 22) recognize that nearly all of the valley walls have talus or debris flow deposits at their base; many (64%; 14 out of 22) recognize the variable density of river terrace risers along the length of the valley."

L185: "Geomorphologically-reasoned" rather than Geologically-reasoned seems more appropriate" in title of section and text. Unless students are relating their geomorphic observations back to the underlying bedrock geology, if so, explain in text and potentially provide an example.

We understand why the reviewer is making this comment but prefer to use "geologically-reasoned" for a few reasons:

(1) We consider geomorphology to be a sub-discipline of geology and prefer the more generalized use here and certainly in our article's title. Our activity's progression from VFT to professional-style decision making is more broadly applicable as a learning strategy to just geomorphology and we wish our readership to consider other ways in which they might be able to adopt a similar progression in their development of learning activities (structural geology, for example).
(2) Students are encouraged to consider the nature of the hazards (floods predictable ~1 week out, rockfalls not predictable; more to do with our forecasting ability than geomorphology) and the facility use (parking lots in flood zones can perhaps be evacuated with limited damage, a seldom-visited storage yard is better near a rockfall hazard zone than a campsite, etc.) when providing their recommendations; we argue these factors extend beyond geomorphology.
(3) It could be argued that identification and mitigation of natural hazards extends beyond the discipline of geomorphology, and that professionally this task is often completed by someone with the title of geologist, not geomorphologist, even when geomorphic data play a considerable role. Many states in the U.S. have Professional Geologist licensures that are a part of a career path for many of our students; these individuals make public-serving decisions regarding natural hazards. Many countries have national "geological surveys" that help with decisions surrounding natural hazards including rockfall and floods. We want the students (and our readers) to more immediately make the connection to geologically-reasoned decisions, rather than geomorphically.
(4) While we point to Part III being a "geomorphic map" exercise we could equally have pointed to it being a "Quaternary geologic map" exercise. The products would largely be the same.

We think the important role of the student's geomorphic mapping and geomorphic knowledge in Part IV of the project is made clear enough in the text. We hope that the reviewer agrees with us that geomorphology is a sub-discipline of geology and that "geologically-reasoned decisions" have a more direct link for student and general public understanding to professional/governmental natural hazard mitigation.

L187: Is hazard 'line' a common term? Consider polygon, outline, or area, unclear what "hazard line" describes as a noun.

To avoid confusion, we have changed references of "hazard lines" throughout the text to "hazard extents".

L199: Would 'Gynomorphically-sound' work better in this context?

No! Having said that we have to assume the reviewer intended to ask whether "geomorphically-sound" would work better. Please see our explanation in response to the comment at Line 185 above.

L219: 'Look direction' informal, consider "view orientation".

As we point to above "Look direction" is commonly used to describe camera positions and satellite view orientations; we thus would not consider this informal but nevertheless have changed the text to "view orientation".

L226: May also be worth mentioning the geology specific virtual outcrop repository V3Geo.

Sure, added.

L271: Link does not work on 17/20/21

Thanks for catching this. The Internet Archive Wayback Machine seems to have last crawled that URL 16 September 2021. At some point since ASU dropped the "www" from their website and did not set the old address to forward. We have updated the link to its current site (once again worrying to see this second instance of a major link change in such a short amount of time).

L289-292: Unclear the relevance of these two sentences, either elaborate and link to the VFT presented or remove.

We have clarified the relevance through rewording the first sentence and deleting the second. The first paragraph now reads: "Undoubtedly a major advantage of VFTs is their immediate accessibility to locations around the globe and beyond. Additionally, VFTs through Google Earth provides opportunities for students to participate in cognitively engaging and interactive learning experiences, which have been found to improve student outcomes in STEM courses (Freeman et al., 2014; National Research Council, 2011). In a recent meta-analysis synthesizing the findings from 225 studies, Freeman and colleagues (2014) found that interactive and active learning experiences, such as those provided by VFTs, are better for students' STEM learning than pedagogical experiences that rely heavily on didactic instruction (e.g. lecture-based instruction)."

L296: What does "excessive cognitive demands" refer to and how is excessive defined?

Reviewer 1 also sought clarification here. What we mean by excessive cognitive demands is extraneous cognitive load. We have revised the paragraph to be clear as to what we mean by extraneous cognitive load and how the Google Earth VFT could potentially decrease students' cognitive load, facilitating learning. This paragraph is now: "VFTs using Google Earth allow for students to engage in guided discovery learning (Mayer, 2004). Virtual learning environments based solely on discovery learning allow students to independently explore and solve problems with little to no guidance (Lee and Anderson, 2013; Mayer, 2004). However, one drawback to this type of learning environment is that it can be a source of extraneous cognitive load, especially when it comes to users with limited knowledge about geology and conducting fieldwork. Cognitive load refers to the load that performing a particular task imposes on the cognitive system (e.g. Paas and Van Merrienboer, 1994; Sweller, 2011). The amount of

information one's cognitive system can process at a given moment is limited. Thus, the presentation of too much information, some of which is unnecessary information when it comes to solving the task, can result in artificially increasing the cognitive resources needed to process the relevant content. This is referred to as extraneous cognitive load by Sweller (2010), and it results in decreasing the efficiency and efficacy of the learner's cognitive system (see review of cognitive load theory by Paas et al., 2010). With the aim of reducing the learner's cognitive load, the embedded instructional prompts that the Google Earth VFT format affords enables the instructor to promote guided discovery of the environment by allowing for the integration of direct instruction into discovery learning (e.g. Lee and Anderson, 2013; Mayer, 2004). This direct instruction provides students with the scaffolding (i.e. additional learning supports that can eventually be decreased with increasing ability) necessary to navigate and learn from such a perceptually and informationally rich environment (e.g. Lee and Dalgarno, 2011)."

L299: Consider "framework" rather than "scaffolding".

Reviewer 1 also was unfamiliar with the use of this term in geoscience education literature. It is an important term that conveys intended meaning in this instance.

Literally, scaffolding is a network of platforms temporarily applied to allow workers to access and construct buildings. The term is pervasive in educational literature since the 1970s. Generally scaffolding is where a teacher provides a lot of support initially to aid students and then progressively removes that support as the student gains confidence and ability.

For the reviewer's sake, here is a useful summary of scaffolding as a method in education: https://www.gcu.edu/blog/teaching-school-administration/what-scaffolding-education

For the sake of readers less familiar with pedagogical terms, we have added a parenthetical after the first in-text appearance that clarifies it as "additional learning supports that can eventually be decreased with increasing ability" and also added a reference to provide further background.

L308: Example using tree unclear, consider another.

We agree and have provided more geologically relevant examples here (the size of the minerals, the fault slightly offsetting the layers, the talus pile at the base of the outcrop).

L301: Again "scaffold", while I understand the use doesn't seem right in this context.

We think Google Earth VFTs are effective learning supports to students' spatial skills and the rest of the paragraph goes on to describe how they can do this. Any other word choice would have less meaning; the word use is precise and we argue appropriate to the context.

L324-328: percentages given but no information on the methodology, numbers of students involved and percentage of class. Please elaborate.

We clarified that 10 students were involved and that the feedback was collected through an anonymous web-survey (i.e. Google Forms). We consider this a small dataset and thus it plays a small role in our study; we do not wish to elaborate further than that, for example, to spell out the exact questions used in the survey and present the supplemental write-in comments.

L335-336: as with upper-division, 'major' also not commonly used external to US to my knowledge.

We have changed this to "advanced undergraduate students" to be more broadly understood.

L345: figure 1. text not legible within B, unclear if this is needed.

Thanks for pointing this out. Illustrating the overall layout is more critical than being able to read the text. Due to the dynamic nature of the VFT format (self-adjusting to fit the screen size), it is problematic taking a screenshot and resizing it for a figure panel. In response to this comment, we reprocessed the screenshot using image sharpening tools and exported as a higher quality (fewer artifacts) and higher resolution (800dpi) image and replaced the previous image in the figure. The text is now legible at high zoom levels.

L359: figure 3: scales on screenshots would benefit readers unfamiliar with the area.

Each panel is a 3D view where the scale varies so we were hesitant to include a scale. However, we agree with this comment and have added a labeled width arrow at a key location in each image to help with scale. We also replaced the screenshots we had with higher resolution images for additional clarity.

L380: Use of "hazard lines", clearer in context of figure than in text as they are further explained here.

At your suggestion we have changed the use of "hazard lines" to "hazard extents".

L445-447: No mention of Putnam et al, 2015 in text.

It was referenced at Lines 143-144. No change.

On behalf of the authors, Nicolas Barth (nic.barth@ucr.edu) 28 October 2021

END